# Enhanced rockwall retreat and modified rockfall magnitudes/frequencies in deglaciating cirques from a 6-year LiDAR monitoring.

Ingo Hartmeyer[1], Markus Keuschnig[1], Robert Delleske[1], Michael Krautblatter[2], Andreas Lang[3], Lothar Schrott[4], Günther Prasicek[3,5], Jan-Christoph Otto[3]

[1]GEORESEARCH Research Institute, Wals, 5071, Austria
[2]Chair of Landslide Research, Technical University of Munich, Munich, 80333, Germany
[3]Department of Geography and Geology, University of Salzburg, Salzburg, 5020, Austria
[4]Department of Geography, University of Bonn, Bonn, 53115, Germany
[5]Interdisciplinary Centre for Mountain Research, University of Lausanne, Bramois, 1967, Switzerland

*Correspondence to*: Ingo Hartmeyer (ingo.hartmeyer@georesearch.ac.at)

**Abstract.** Cirque erosion contributes significantly to mountain denudation and is a key element of glaciated mountain topography. Despite long-standing efforts, rates of rockwall retreat and the proportional contributions of low-, mid- and high magnitude rockfalls have remained poorly constrained. Here, a unique, terrestrial LiDAR-derived rockfall inventory (2011-2017) of two glaciated cirques in the Hohe Tauern Range, Central European Alps, Austria is analysed. The mean cirque wall retreat rate of 1.9 mm a$^{-1}$ ranks in the top range of reported values and is mainly driven by enhanced rockfall from the lowermost, freshly deglaciated rockwall sections. Retreat rates are significantly elevated over decades subsequent to glacier downwasting. Elongated cirque morphology and recorded cirque wall retreat rates indicate headward erosion is clearly outpacing lateral erosion, most likely due to the cataclinal backwalls, which are prone to large dip-slope failures. The rockfall magnitude-frequency distribution – the first such distribution derived for deglaciating cirques – follows a distinct negative power law over four orders of magnitude. Magnitude-frequency distributions in glacier-proximal and glacier-distal rockwall sections differ significantly due to an increased occurrence of large rockfalls in recently deglaciated areas. In this paper, the second of two companion pieces, we show how recent climate warming shapes glacial landforms, controls spatiotemporal rockfall variation in glacial environments and indicates a transient signal with decadal scale exhaustion of rockfall activity immediately following deglaciation crucial for future hazard assessments.

## 1 Introduction

Cirque erosion contributes significantly to the morphological appearance of glaciated mountain ranges. It controls rockwall retreat and creates emblematic high-alpine landform features such as horn-type peaks and sharp-edged ridges. Erosional processes operating in glacial cirques are widely recognized as important agents of high-alpine landscape evolution (Benn and Evans, 2010; Sanders et al., 2012; Scherler, 2014). Rockfall from cirque walls represents a primary source of debris for glacial

systems and thus supplies tools for effective glacial erosion (Hallet, 1981). The disposal of sediment from cirque walls also represents a prominent entry point to the high-alpine sediment cascade and is, therefore, key to understanding of high-alpine sediment flux (Hales and Roering, 2005; Krautblatter et al., 2012; Bennett et al. 2014).

Cirque wall retreat is governed by rock slope failure, which can be statistically characterized by their magnitude-frequency

distribution (Dussauge et al., 2003; Bennett et al., 2012). Magnitude-frequency distributions are widely used to derive probabilistic recurrence rates of an event of a given size (Dussauge-Peisser et al., 2002) and are key to understanding process efficiency. Mass movement size-distributions can usually be described by a power law (Hovius et al., 1997). Magnitude-frequency distributions of rockfall reflect rock mass properties and triggering mechanisms, and therefore change over time (Krautblatter and Moore, 2014). They constitute an important tool for understanding headwall geomorphology (Densmore et

al., 1997), slope response to climatic and environmental change (Schloegel et al., 2011), sediment transport rates (Korup, 2005) and hazard potential (Krautblatter and Moser, 2009). Elevated rockfall activity from freshly excavated rockwalls, such as investigated here, is of major concern in all high-mountain regions, particularly where human pressure increases (Fischer et al., 2011; Purdie, 2013). Due to continuing climate warming its importance is expected to increase throughout the foreseeable future, making accurate knowledge of magnitude-frequency distributions essential for effective risk assessment throughout

mountain regions and safeguarding high-alpine infrastructure (Arenson et al., 2009; Bommer et al., 2010).

High-alpine cirques were first studied in the early 20th century when cirque walls were considered to wear back through sapping at the base of the headwall and inside the Randkluft (gap between glacier and headwall) through intense frost weathering (Richter, 1900; Martone, 1901; Johnson, 1904). Subsequently, determining ground thermal conditions at the headwall base underlined the importance of periglacial weathering for cirque wall retreat (Gardner, 1987; Sanders et al., 2012).

Additional process like rotational ice-flow (Lewis, 1949; Waldrop, 1964), enhanced quarrying due to subglacial meltwater drainage (Hooke, 1991; Iverson, 1991), strong abrasion under thick ice (Strøm, 1945), and slope collapse (Evans, 1997) are also considered key agents of cirque expansion. Numerous morphometric studies have focused on cirque shape and found remarkably similar cirque length and width across a number of mountain ranges (see review in Barr and Spagnolo, 2015). Deviations from the typical circular cirque-planform were mainly encountered where isometric cirque growth is modified by

geological structure (Bennett and Glasser, 2009; Evans, 2006). Despite an extensive research focussing on cirques, cirque erosion is still poorly constrained and the number of published erosion rates has remained limited. Cirque wall retreat rates have been quantified using a variety of different approaches including long-term averages based on sediment deposits (e.g. Larsen and Mangerud, 1981), cosmogenic dating (e.g. Heimsath and McGlynn, 2008), and cirque allometry (e.g. Evans et al., 2006) as well as short-term monitoring studies based on lacustrine deposits (e.g. Hicks et al., 1990), supraglacial scree (e.g.

O'Farrell et al. 2009), and terrestrial LiDAR (e.g. Kenner et al., 2011).

Inventories of rockfall and rock slope failures in high-alpine environments cover a wide range of spatial and temporal scales. Studies typically focus on catchment scale (Cossart et al., 2008; Krautblatter et al., 2012) or orogen scale (Noetzli et al., 2003; Allen et al., 2011) and mid- to high-magnitude events. Methods used to compile inventories are diverse and include field mapping and aerial photograph interpretation (Holm et al., 2004; Fischer et al., 2012), information from observer networks

(Ravanel et al., 2010), comparisons of historical and recent photographs (Ravanel and Deline, 2010), direct observations (Fischer et al., 2006), and digital photogrammetry from aerial photographs (Bennett et al., 2012).

Comprehensive quantitative studies that also consider low-magnitude events and thus comprise the entire rockfall spectrum are rare. First approaches date back to the middle 20th century (Rapp, 1960). Over the last two decades, the emergence of LiDAR led to a marked increase of quantitative, high-resolution rockfall studies (Derron and Jaboyedoff, 2010) and provided

new insights on rockfall magnitude-frequency distributions (Abellán et al., 2011). However, due to limited resolution LiDAR systematically undersamples the smallest magnitudes (rockfall particles < 10 cm) (Lim et al., 2010), which in some geological settings contribute significantly to rock slope mass wasting (Krautblatter et al., 2012).

Detailed multiannual LiDAR inventories to investigate magnitude-frequency distributions of rockfall were compiled for coastal cliffs (Rosser et al., 2005; Lim et al., 2006), volcanic rock slopes (Nguyen et al., 2011), granitic and limestone rockwalls

(Stock et al., 2011; Strunden et al., 2015), and in urban areas (Abellán et al., 2011). Such studies have focussed on precursory events and pre-failure deformation (Rosser et al., 2007; Abellán et al., 2010), disintegration monitoring of single rock slope failures (Oppikofer et al., 2008; Oppikofer et al., 2009), evaluation of rock masses near infrastructure (Hungr et al., 1999; Lato et al., 2012) and analysis of long-term rockwall retreat (Heckmann et al., 2012; Strunden et al., 2015).

For glaciated cirques, no rockfall magnitude-frequency distributions have been reported so far. Other steep bedrock

environments that have been targeted are coastal cliffs (Williams et al., 2018), small debris flow basins (Bennett et al., 2012), gorge systems (Dussauge-Peisser et al., 2002) and valley flanks (Guzzetti et al., 2003). Rockfall studies in cirque environments either focused on large singular events (Rabatel et al., 2008), on rockfall around infrastructures (Ravanel et al., 2013), or on spatially limited sections of (deglaciating) cirques (Kenner et al., 2011). Spatiotemporal rockfall patterns and rockwall retreat rates for entire cirques over several years have not yet been reported. Despite research efforts reaching back more than 100

years, contemporary cirque wall retreat rates and the efficiency of specific magnitudes have remained unconstrained representing a significant problem for landscape evolution studies and a challenge for rockfall risk management (Brocklehurst and Whipple, 2002; Benn and Evans, 2010; Scherler, 2014).

Here we address this research gap by analysing a comprehensive rockfall inventory compiled during six years (2011-2017) of repeat laserscanning of the same rockwalls in two neighbouring, glacierized cirques in the Hohe Tauern Range, Central Alps,

Austria. We (i) identify significantly increased cirque wall dismantling in the vicinity of the current glacier surface, (ii) observe mean cirque wall retreat rates that rank among the highest values reported yet, (iii) reveal that headward erosion outpaces lateral erosion consistent with present cirque shape, and (iv) quantify the first ever magnitude-frequency distribution from a deglaciating cirque. Results are framed in the existing knowledge on magnitude-frequency relationships and considering effects of continuing climatic changes. This paper is closely linked to a companion paper (Hartmeyer et al., 2020) which

identified significant glacial thinning (0.5 m a$^{-1}$) adjacent to the monitored rockwalls and found elevated rockfall activity in the freshly deglaciated terrain. 60 % of the rockfall volume detached from less than ten vertical meters above the glacier surface. High rates 10-20 m above the glacier indicate enhanced rockfall activity over tens of years following deglaciation. Rockfall preconditioning probably starts inside the Randkluft (void between cirque wall and glacier) where sustained freezing

and ample supply of liquid water likely causes enhanced physical weathering and high plucking stresses. As the glacier is wasting down strong temperature variations will induce pronounced thermal stress in the first-time exposed rock, cause rock fatigue and lead to the formation of a deep active layer, all of which will exert significant destabilizing effects in glacier-proximal areas.

## 2 Study Area

Two neighbouring glacial cirques at the Kitzsteinhorn, Central European Alps, Austria (Fig. 1) were chosen as study site. The two north-facing cirques (~ 0.3 km²), referred to as eastern cirque and western cirque, constitute the root zone of the Schmiedingerkees glacier (~ 0.8 km²), which has retreated considerably in recent years. Currently, the receding ice masses are constraint within the cirques, a state that is characteristic for many glaciers in the Eastern European Alps (Fischer et al., 2018). Adjacent to the investigated rockwalls, warming-related glacial thinning has led to downwasting rates around 0.5 m per year over the last decade leading to the exposure of fresh cirque wall sections (Hartmeyer et al., 2020). Mean annual air temperature and annual precipitation, both measured at a local weather station situated on the upper part of the Schmiedingerkees glacier (2,940 m a.s.l.), vary around -2 °C and 2,500 mm, respectively. Local borehole temperature monitoring demonstrates the existence of permafrost with temperatures of -1.8 °C (north-facing) and -1.3 °C (west-facing) at zero annual amplitude depth.

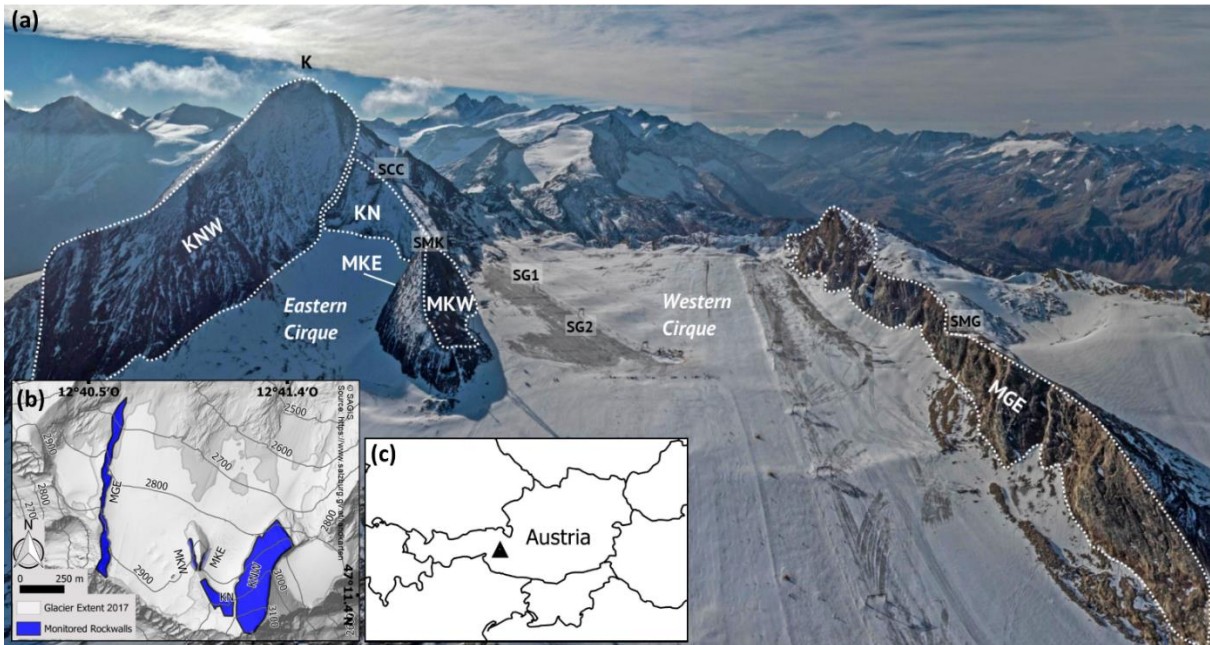

**Figure 1: UAV view of rockwalls investigated (surface area, elevation above sea level, mean gradient are indicated in brackets): KN (23,500 m²; 2,920-3,060 m a.s.l.; 47°), KNW (133,400 m²; 2,770-3,200 m a.s.l.; 44°), MKE (11,100 m²; 2,850-2,950 m a.s.l., 71°), MKW (6,300 m²; 2,880-2,940 m a.s.l.; 63°), MGE (60,400 m²; 2,740-2,990 m a.s.l.; 61°) (Photo: Robert Delleske). 1b: Hillshade of study area with monitored rockwalls (blue) and glacier extent. 1c: Location of study area within Austria. Abbreviations: K = Kitzsteinhorn (Summit), SMK = Scan Position 'Magnetkoepfl', SCC = Scan Position 'Cable Car Top Station', SG1 = Scan Position 'Glacier 1', SG2 = Scan Position 'Glacier 2', SMG = Scan Position 'Maurergrat', see text for further abbreviations (Photo: UAV/R. Delleske).**

The Schmiedingerkees glacier and the immediately adjacent summit pyramid of the Kitzsteinhorn constitute one of Austria's most frequented high-alpine tourist destinations with close to one million visitors per year. The Kitzsteinhorn hosts an extensive research site to investigate the consequences of climate change on high-alpine infrastructure and rock stability ('Open-Air-Lab Kitzsteinhorn'). Measurements performed at the Kitzsteinhorn focus on rockfall activity (Keuschnig et al., 2015), subsurface temperature changes (Hartmeyer et al., 2012), geophysical monitoring with ERT (Supper et al., 2014;

Keuschnig et al., 2016), rock mass pressure using anchor load plates (Plaesken et al., 2017) and fracture dynamics monitoring with crackmeters (Ewald et al., 2019).

Full details of the study site are presented in Hartmeyer et al., 2020. In brief, the total surface area of the investigated rockwalls is 234,700 m² and their mean vertical extent ranges between 35-200 m. All studied cirque walls are immediately adjacent to the Schmiedingerkees glacier: In the eastern cirque the Kitzsteinhorn north-face (KN) (backwall), Kitzsteinhorn northwest-

face (KNW) and Magnetkoepfl east-face (MKE) (sidewalls). In the western cirque no significant backwall exists and the Magnetkoepfl west-face (MKW) and Maurergrat east-face (MGE) were monitored (Fig. 1).

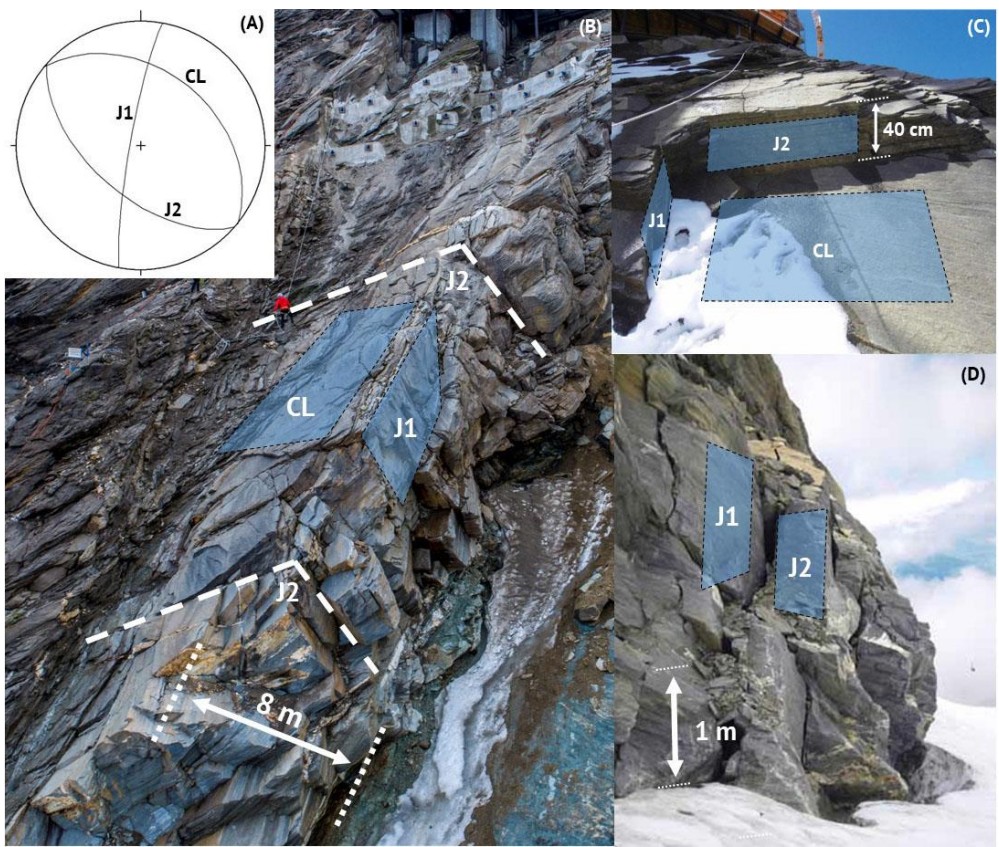

**Figure 2: Dip and strike of the dominant discontinuities at the studied rockwalls (Fig. 2A). Images of Kitzsteinhorn north-face (KN) (Fig. 2B/C) and Magnetkoepfl east-face (MKE) (Fig. 2D) with cleavage (CL) and joint sets (J1, J2) indicated. Cleavage of the**
**calcareous mica-schists dips about 45° NNE. Joint sets J1 (dipping subvertical to W) and J2 (dipping steeply to SW) are approximately orthogonal to CL and predispose north-facing slopes for dip-slope failures (Fig. 2B/C), while west- and east-facing areas are more susceptible to toppling failures (Fig. 2D) (Photos: R. Delleske (2B), A. Schober (2C, 2D)).**

All cirque walls developed in rocks of the Bündner schist formation within the Glockner Nappe and belong to the Glockner Facies consisting of calcareous micaschist, prasinite, amphibolite, phyllite, marble and serpentinite (Cornelius and Clar, 1935; Hoeck et al., 1994). Within the monitored rockwalls, NNE-dipping (~ 45 °) calcareous micaschists dominate, and isolated marble and serpentinite belts exist at Magnetkoepfl. Two distinct joint sets (J1 dipping steep to W, J2 medium-steep to SW) oriented approximately orthogonal to the cleavage precondition disintegration into cubic rock fragments (Fig. 2). Numerous open fractures infilled with fine-grained material enable water infiltration and affect near-surface rock slope kinematics and thermal dynamics (Keuschnig et al., 2016; Ewald et al., 2019). Rock at the surface is highly fractured due to a pronounced frost weathering susceptibility. Strong tectonic forcing resulted in highly fractured weakness zones along major faults in all rockwalls. Rock mass classifications according to Romana (1985) and Bieniawski (1993) suggest highly variable lithological strength ranging from low stability values in weakened zones (r = 34) and highly stable conditions in steep, unweathered sections (r = 98) (Terweh, 2012).

## 3 Methods

### 3.1 LiDAR Data Acquisition and Processing

Full details of data acquisition can be found in Hartmeyer et al., 2020. In brief, terrestrial LiDAR acquisition was based on a Riegl LMS-Z620i laserscanner equipped with a calibrated high-resolution digital camera to take referenced colour images. Rockwalls were scanned in reflectorless mode at least once a year between 2011 and 2017 at the end of the ablation period. A total of 56 rockwall scans were carried out with variable mean object distances (140-650 m) depending on the used scan position. Resulting point cloud resolution typically ranges between 0.1-0.3 m (see supplement of Hartmeyer et al., 2020 (Table S2) for a full list of data acquisition parameters).

Iterative-Closest-Point (ICP) procedures (Chen and Medioni, 1992) were applied for point cloud alignment. Areas that have been subject to surface change between surveys were discarded during the process and therefore do not negatively affect the quality of surface matching (Abellán et al., 2010; Abellán et al., 2011), resulting in alignment errors between 1.5-3.7 cm. Surface changes between point clouds were identified using the M3C2 algorithm of Lague et al., 2013 due to is robust performance on irregular surfaces, with missing data and changes in point density. For each distance measurement the algorithm calculates the local confidence interval (at one sigma level), which was added to the alignment error and propagated into the volume error. Full details on rockfall volume computation and error quantification are provided in Hartmeyer et al., 2020. In addition to rockfall volume and its associated uncertainty a suite of morphometric parameters including slope aspect, gradient and elevation above glacier surface was determined for each rockfall source area.

### 3.2 Rockfall Magnitude-Frequency Calculation

Numerous studies (e.g. Hungr et al., 1999) demonstrated that the relationship between rockfall volume and cumulative rockfall frequency can be defined by a power law following Eq. (1):

$$f(V) = \alpha V^{-b} \tag{1}$$

where $f(V)$ is the cumulative number of rockfalls, $V$ is the rockfall volume, $\alpha$ is the pre-factor, and $b$ is the power law exponent. To test the power-law-fit of an empirical distribution, earlier studies mainly relied on two regression approaches: cumulative distribution functions (CDF) and probability density functions (PDF) (Bennett et al., 2012; Strunden et al., 2015). According to (Bennett et al., 2012) the PDF is better suited to visualize rollovers, i.e. a decrease in the frequency density for small events. It requires, however, logarithmic binning of rockfall volumes that is rather subjective and has been demonstrated to introduce

bias in the calculation of power law exponents (Clauset et al., 2009; Bennett et al., 2009). As will be shown in Sect. 4.4, the magnitude-frequency data used here does not show a rollover at low event sizes and so CDFs were constructed using the R package poweRlaw (Gillespie, 2015).

Following earlier analyses of rockfall magnitude-frequency distributions (Barlow et al., 2012; Strunden et al., 2015) the sensitivity of the power law exponent was tested using a bootstrapping approach (Monte Carlo simulation), in which 20 % of

the rockfalls were randomly removed and the data set resampled 100,000 times to calculate median, 2.5th and 97.5th percentiles.

### 3.3 Rockwall Retreat Rate Calculation

First, the total rockfall volume registered was divided by the number of observation years to obtain mean annual volume (m³). Second, the volume was divided by the surface area of the investigated rockwall (m²) to derive the (slope-perpendicular)

rockwall retreat rate. Rockwall surface area calculations were carried out in *CloudCompare*: point clouds of rockwalls were first subsampled (thinned) to a homogenous point density of 0.5 m⁻¹ in order to prevent a potential bias due to variable resolution within point clouds. The subsampled point cloud was then used to generate a mesh based on a Delaunay triangulation (maximum edge length 3 m), which served as basis for the surface area calculation.

### 4 Results

**4.1 Rockfall Magnitudes**

Over the course of the monitoring program which investigated the same five cirque walls for six consecutive years (2011-2017), scan positions and acquisition resolution had to be altered resulting in differing data resolution between scans. To enable direct comparison of scans of differing resolution the impact of scan resolution on the number of events detected was constrained. Using a regression analysis it is shown that for events larger than 0.1 m³ scan resolution has no statistically

significant impact (Hartmeyer et al., 2020).

Above this threshold, 374 rockfalls were identified resulting in a total volume of 2,551.4 ± 136.7 m³. Magnitude-frequency distributions of rockfalls often follow a power law function (as is also the case here, see Sect. 4.4). To classify rockfall volumes, the recorded events were grouped into bins of logarithmically increasing size to balance against strongly uneven event volumes

(Fig. 3). This follows the volumetric classification introduced by Whalley (1974, 1984) (debris falls < 10 m³; boulder falls 10-
$10^2$ m³; block falls $10^2$-$10^4$ m³), which is commonly used in science and engineering (e.g. Brunetti et al., 2009; Krautblatter et al., 2012; Sellmeier, 2015).

A dominance of large events is evident as two thirds (67 %) of the rockfall volume fall into the largest size class (100-1,000 m³), while the next smaller class (10-100 m³) accounts for approximately one fourth (23 %), and the two smallest classes (0.1-10 m³) combined constitute only about one tenth (10 %) of the total rockfall volume.

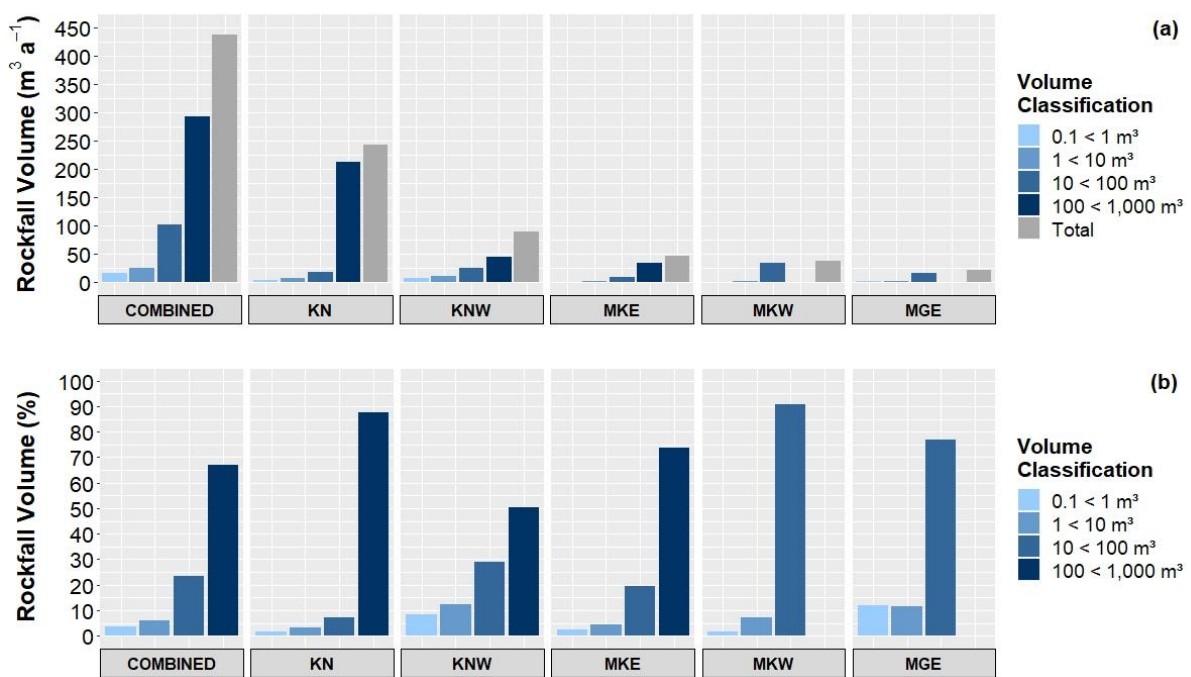


**Figure 3: (a) Absolute rockfall volume (m³ a⁻¹), (b) relative rockfall volume (%) for all monitored rockwalls. About two thirds (67 %) of the total rockfall volume fall into the largest size class (100-1,000 m³) reflecting the dominance of large events.**

When analysed individually for each rockwall, the size distribution shows significant differences due to the impact of individual large rockfalls: At all walls most of the rockfall volume is represented by the largest size classes (at KN rockfalls >
100 m³ contribute 88 %, at MKE and KNW contributions equal 74 % and 50 %, respectively and no rockfalls > 100 m³ occurred at MKW and MGE). The combined share of the two largest size classes (i.e. rockfalls > 10 m³) ranges between 77 % (MGE) and 95 % (KN), whereas the smallest size class (0.1 – 1 m³) represents only around 2 % of the total volume at KN and at the Magnetkoepfl (MKE, MKW) and about 10 % at MGE and KNW.

In a companion study, we demonstrated significantly increased rockfall in the immediate surroundings of the glacier resulting
from antecedent rockfall preparation inside the Randkluft and recent glacial downwasting adjacent to the investigated cirque walls (Hartmeyer et al., 2020). This is also evident if the volume classification is further differentiated according to distance from glacier (Fig. 4): At 0-10 vertical meters above the glacier surface ("proximal areas"), rockfalls larger than 100 m³

constitute 84 % of the total volume whereas in rockwall sections located > 10 m above the glacier surface ("distal areas") the contribution of this class drops to just 41 %. Small rockfalls below 1 m³ on the other hand contribute around 1 % to the total rockfall volume in proximal areas, while in distal areas this part increases to 7 %.

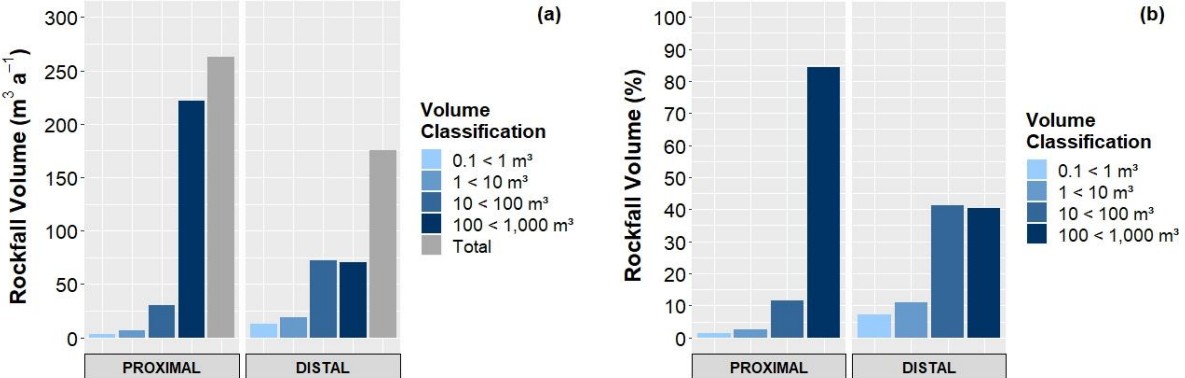

**Figure 4: (a) Absolute rockfall volume (m³ a⁻¹), (b) relative rockfall volume (%) for rockwall sections 0-10 vertical meters above the glacier surface ('Proximal') and for higher (> 10 m) rockwall sections ('Distal'). Large events (100-1,000 m³) dominate rockfall size distributions in proximal areas while distal areas show a more balanced distribution.**

## 4.2 Rockwall Retreat Rates

Across the full six-year period of observation and across all rockwalls the total rockfall volume is 2,551.4 ± 136.7 m³ which corresponds to a mean retreat rate is 1.86 ± 0.10 mm a⁻¹. In detail, annual rockwall retreat varies considerably between the rockwalls investigated and is highest along highly fractured weakness zones close to the glacier surface (Hartmeyer et al., 2020). The maximum rate is found at KN (10.32 ± 0.22 mm a⁻¹), followed by the two rockwalls of the Magnetkoepfl. At MKW retreat rates equal 5.94 ± 0.38 mm a⁻¹, and at MKE 4.20 ± 0.29 mm a⁻¹ was recorded. Lowest rates were calculated for KNW (0.68 ± 0.07 mm a⁻¹) and MGE (0.35 ± 0.04 mm a⁻¹) (Fig. 5).

In proximal areas, mean retreat rates of 7.56 ± 0.22 mm a⁻¹ are found and in distal areas rates are almost an order of magnitude lower (0.88 ± 0.08 mm a⁻¹). By far the highest retreat rates in proximal areas were registered at KN (57.32 ± 0.67 mm a⁻¹), followed by MKE (16.97 ± 1.16 mm a⁻¹). At KN several large rockfalls were recorded adjacent to the glacier surface along a prominent joint intersection. Following the cleavage direction at MKE a ledge of highly fractured micaschists runs diagonally through the rockwall. Here, the highest instability occurs especially in the immediate vicinity of the glacier surface from where a major share of the rockfall volume originates.

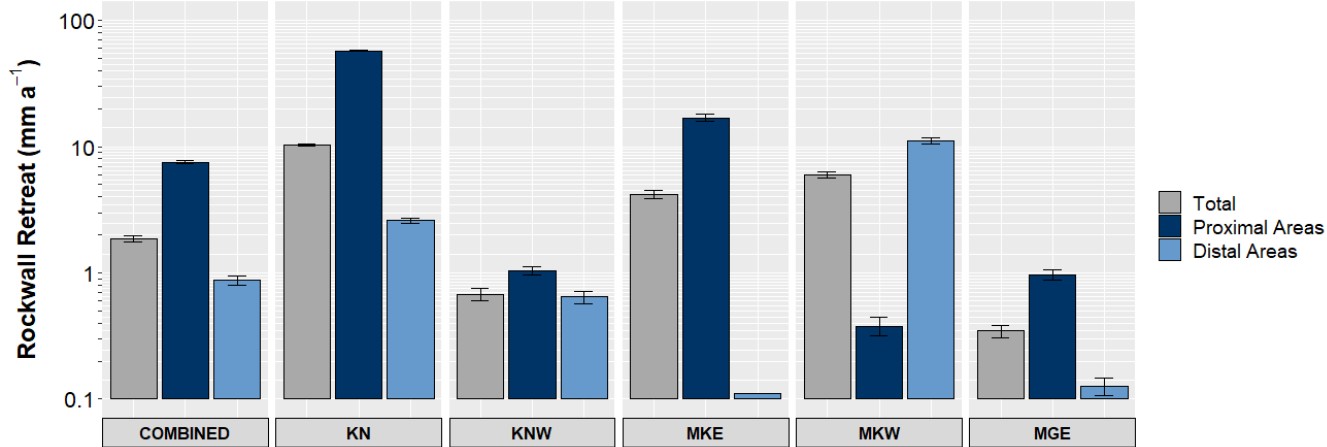

**Figure 5: Rockwall retreat rates (mm a⁻¹) (i) for the entire rockwall (dark red), (ii) for proximal areas (0-10 m, red) and (iii) for distal areas (> 10 m, orange). The mean rate over the six-year observation period is just under 2 mm a⁻¹ were found. Rockwall retreat rates in proximal areas (7.6 mm a⁻¹) were almost an order of magnitude higher than in distal areas (0.9 mm a⁻¹).**

The highest distal retreat rates are found at MKW (11.14 ± 0.67 mm a⁻¹). This is also the only site where retreat rates in the first 10 m are exceeded by retreat rates in more distal sections. due to continuing mass wasting from a rockfall scarp initiated

in the 2000s and located about 15 m above the current glacier surface.

### 4.3 Rockfall Frequencies

To compare rockfall frequency across rockwalls, rockfall numbers were normalized for rockwall size (Fig. 6). Averaged over all five monitored rockwalls and the entire six-year monitoring period, 2.7 rockfalls per 10,000 m² a⁻¹ were registered. Maximum numbers were recorded at KN (n = 7.4) followed by MKW (n = 6.0) while the lowest numbers were found at MGE

(n = 1.5).

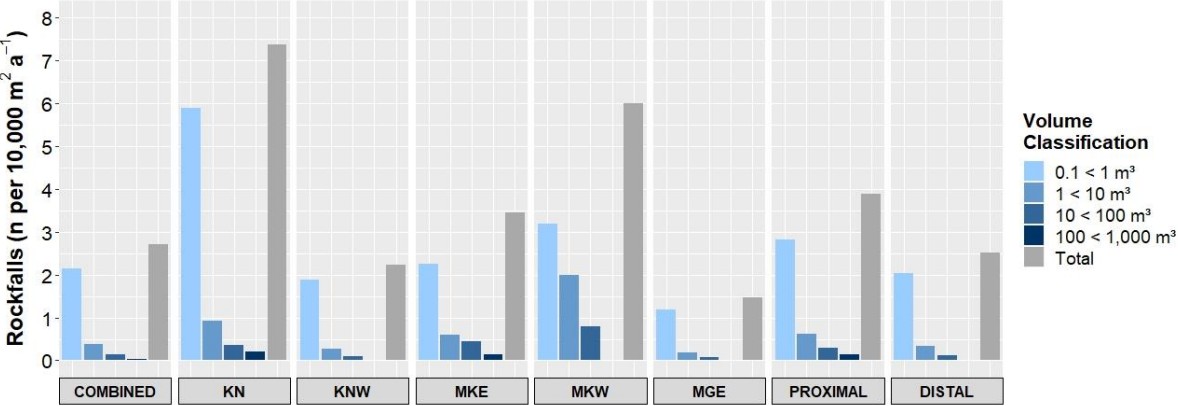

**Figure 6: Normalized rockfall numbers (n per 10,000 m² a⁻¹) for all monitored rockwalls. Rockfall frequency decreases exponentially with increasing rockfall size, 80 % of all rockfalls rank in the smallest size class (0.1-1 m³).**

Rockfall frequency follows a power-law decrease with increasing rockfall size. Almost 80 % of all rockfalls fall in the smallest
size class (0.1-1 m³) while the next larger size class (1-10 m³) comprises 14 % and the two largest size classes (10-100 m³, 100-1,000 m³) 6 % and 1 %, respectively. The frequency pattern is rather similar for all rockwalls, and only the Magnetkoepfl rockwalls (MKE, MKW) show some minor deviations. Here, the smallest size class represents a relatively low share (65 % and 53 %, respectively), whereas the larger size classes account for a comparatively high proportion (see supplement of Hartmeyer et al., 2020 (Table S3) for full details).

The difference between rockfall frequency in proximal areas (n = 3.9) and distal areas (n = 2.5) is considerable yet less pronounced than the (eightfold) difference between proximal (7.6 mm a$^{-1}$) and distal (0.9 mm a$^{-1}$) rockwall retreat rates. Proximal rockfall frequency exceeds distal rockfall frequency for all size classes. The discrepancy between proximal and distal rockfall frequency grows with rockfall size. Rockfalls > 10 m³ constitute 10 % of all proximal rockfalls, where as in distal areas only around 5 % of all rockfalls exceed 10 m³. Rockfalls > 100 m³ represent around 4 % of all proximal rockfalls (< 1 %
in distal areas) and thus occur 8.6 times more often in proximal areas than in distal areas. Low magnitudes represent a smaller share in proximal areas (73 %) and a larger share in distal areas (82 %).

## 4.4 Magnitude-Frequency Distributions

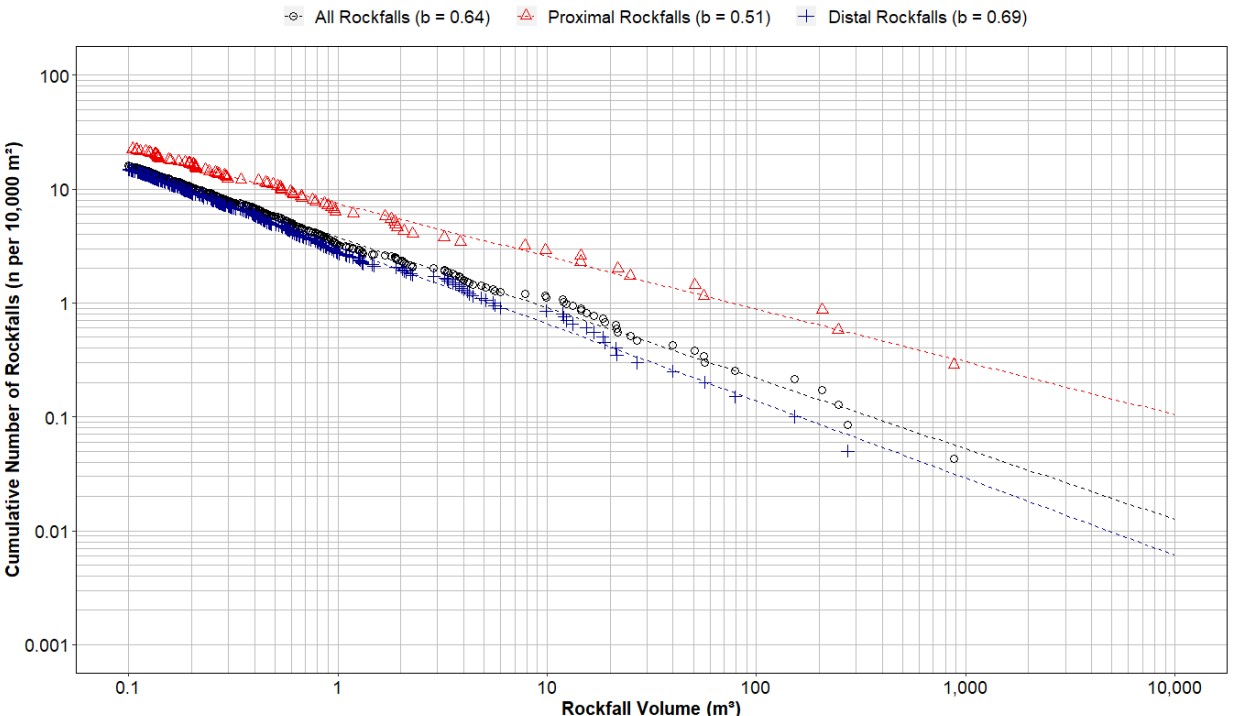

**Figure 7: Magnitude-frequency distribution for (i) all detected rockfalls, (ii) proximal rockfalls and (iii) distal rockfalls. The power**
**law exponent (b) determines the slope of the regression line (straight line in log-log space). Proximal rockfalls (b = 0.51) are fitted by a flatter regression line than distal rockfalls (b = 0.69) pointing to an increased frequency of large rockfalls in the vicinity of the glacier (i.e. in areas more recently exposed by glacier retreat).**

To further characterize spatiotemporal rockfall variations, the magnitude-frequency distributions were fitted using a power-law. The resulting distinct negative power law distribution extends over four orders of magnitude and tails off towards high magnitudes. Including all events the power law exponent is $b = 0.64$ (Fig. 7). Rockfall magnitude-frequency distributions for proximal rockfalls yield an exponent $b = 0.51$ and for distal rockfalls the exponent $b = 0.69$. This significant difference between proximal and distal rockfall magnitude-frequency distributions is primarily caused by an increased frequency of large rockfalls in proximal rockwall sections.

Accounting for the stochastic nature of rockfall processes, the robustness of the power-law-fit was estimated by a bootstrapping simulation. Results demonstrated only minimal variations of the power law exponent $b$ for the entire ($0.64^{+0.04}_{-0.03}$), proximal ($0.51^{+0.07}_{-0.05}$), and distal dataset ($0.69^{+0.04}_{-0.03}$) at a 95 % confidence interval. To selectively examine the sensitivity of the regression to individual rare events, the power-law-fits were recalculated after omitting the five largest events (rockfalls > 100 m³) from the dataset. The results show only minor deviations with slightly increased power law exponents (+0.03) for the magnitude-frequency distribution using the reduced dataset (+0.08 for proximal rockfalls, +0.02 for distal rockfalls). The large rockfalls recorded (100-1,000 m³) do not fundamentally alter the regression underlining the robustness of the magnitude-frequency relationships.

Ignoring lithological and topographic constraints and extrapolating the inferred power law beyond the upper end of the observed magnitude spectrum (~ 1,000 m³) allows theoretical estimates of recurrence intervals of high-magnitude events. For rockfalls ≥ 1,000 m³ a power-law-predicted return period of around four years is indicated while rockfalls ≥ 10,000 m³ (≥ 100,000 m³) recur approximately once every 25 years (100 years). With increasing rockfall magnitude proximal and distal regression lines increasingly diverge. Normalized for rockwall size, rockfalls ≥ 1,000 m³ are around ten times more likely in proximal rockwall sections than in distal rockwall sections and about 15 times (25 times) more likely for rockfalls ≥ 10,000 m³ (≥ 100,000 m³) (Fig. 7).

## 5 Discussion

In the present study five cirque walls were monitored with terrestrial LiDAR for six consecutive years (2011-2017). In the ensuing section we relate our findings to similar studies carried out in high-alpine environments and discuss potential implications for erosional processes in cirques and magnitude-frequency relationships in a changing climate.

### 5.1 Cirque Erosion

Erosional processes are both, highly discontinuous and unsteady over time (Sadler, 1981), particularly in paraglacial environments (Ballantyne, 2002; McColl, 2012). Erosion rates recorded during short observation periods are therefore unlikely to accurately reflect long-term rockwall retreat over geological time scales (Krautblatter et al., 2012). High-magnitude events occur with lower frequency and are usually undersampled, contributing to a systematic underestimation of long-term rockwall retreat (e.g. Strunden et al., 2015). Observations made during periods of elevated geomorphic activity result in an

overestimation of long-term rates (e.g. Ballantyne and Benn, 1994). Interpretation of short-term erosion records in a broader landscape evolution context thus requires careful consideration of the respective geomorphological setting.

To date, only few retreat rate estimates have been reported for cirques (see reviews in Benn and Evans, 2010; Sanders et al., 2013; Barr and Spagnolo, 2015). Investigations carried out in cirques in the Sangre de Cristo Range, USA (Grout, 1979), Kråkenes, Norway (Larsen and Mangerud, 1981), the Ben Ohau Range, New Zealand (Brook et al., 2006), the Annapurna Range, India (Heimsath and McGlynn, 2008), and the Rocky Mountains, Canada (Sanders et al., 2013) all report (long-term) rockwall retreat rates around or below 1 mm a$^{-1}$ (Tab. 1). Higher rates (> 2 mm a$^{-1}$) were reported for headward cirque retreat in soft volcanic breccia in Antarctica (5.8 mm a$^{-1}$) (Andrews and LeMasurier, 1973), for selected sections of a deglaciating rocky ridge in the Swiss Alps (6.5 mm a$^{-1}$) (Kenner et al., 2011), and for cirque-wall sections in the French Alps affected by a single, large rockfall event (8.4 mm a$^{-1}$) (Rabatel et al., 2008). These results were, however, not derived from cirque-scale monitoring but instead reflect short-term, local erosion rates in selected, highly active rockwalls. Such data is better compared to the range of maximum headward erosion rates established at the Kitzsteinhorn, the maximum of which is 10.3 mm a$^{-1}$ at KN.

**Table 1: Comparison of published cirque wall retreat rates (extended from Sanders et al., 2013; Barr and Spagnolo, 2015)**

| Location | Retreat rate (mm a$^{-1}$) | Time scale (years) | Object investigated | Method | Reference |
|---|---|---|---|---|---|
| Kråkenes, Norway | 0.1 | 700 | Cirque | Glaciolacustrine sediments | Larsen and Mangerud (1981) |
| Ben Ohau Range, New Zealand | 0.4 | 7.5 x 10$^5$ | Single headwall | Space for time | Brook et al. (2006) |
| Principal Cordillera, Argentina | 0.9-1.1 | 10$^4$ | Cirque | Debris volume of rock glacier | Schrott (1996) |
| Sangre de Cristo Range, USA | 0.9-1.3 | 10$^4$ | Cirque | Debris on rock glacier | Grout (1979) |
| Rocky Mountains, Canada | 1.2 | 50 | Cirque | Rockfall monitoring (LiDAR) | Sanders et al. (2013) |
| Annapurna Range, India | 1.3 | 10$^2$-10$^3$ | Headwall | Supraglacial sediments | Heimsath and McGlynn (2008) |
| Marie Byrd Land, Antarctica | 5.8 | 3-8 x 10$^5$ | Headwall | Landform change | Andrews and LeMasurier (1973) |
| European Alps, Switzerland | 6.5 | 4 | Headwall | Rockfall monitoring (LiDAR) | Kenner et al. (2011) |
| European Alps, France | 8.4 | 1 | Headwall | Rockfall monitoring (LiDAR) | Rabatel et al. (2008) |
| Present study | 1.9 | 6 | Cirque | Rockfall monitoring (LiDAR) | - |
| Present study | 10.3 | 6 | Headwall | Rockfall monitoring (LiDAR) | - |

Direct cross-study comparison across various spatiotemporal scales is difficult due to differing study designs and key environmental factors (Barr and Spagnolo, 2015). Still, the mean retreat rate at the Kitzsteinhorn of just under 2 mm a$^{-1}$ is one

of the highest values published worldwide. Recent extensive compilations of mostly long-term rockwall retreat rates from periglacial environments (not restricted to cirques) yield similar results, with rates generally below 1 mm a$^{-1}$ (de Haas et al., 2015; Ballantyne, 2018). Taking it beyond Earth, Kitzsteinhorn results still are at the higher end of a range of time-span corrected terrestrial and extraterrestrial rockwall retreat rates as compiled by de Haas et al., 2015 (see their figure 12 and table

S1). Higher rates were reported for headwalls above rock glaciers in the Swiss Alps (2.5 mm a$^{-1}$) (Barsch, 1977), the French Alps (2.5 mm a$^{-1}$) (Francou, 1988) and West Greenland (5 mm a$^{-1}$) (Humlum, 2000), and for rock slopes subject to intense chemical weathering (2.5-6 mm a$^{-1}$) (Åckerman, 1983; Dionne and Michaud, 1986).

Long-term cirque evolution is considered to show highest erosion rates during the transition to ice-free conditions, when cirques are occupied by small glaciers only. Cirque growth is thus considered being far from a steady process but meant to

happen in spurts during deglacial and interglacial periods (Delmas et al., 2009; Crest et al., 2017). The data analysed here and in a companion study (Hartmeyer et al., 2020) confirms such patterns and reveals considerably elevated retreat rates in recently deglaciated glacier-proximal areas (7.6 mm a$^{-1}$), while more distal rockwall sections show retreat rates of less than 1 mm a$^{-1}$ (Fig. 5). High rates in proximal areas are directly related to glacier downwasting, which is expected to induce pronounced thermal stress, cause rock fatigue through cyclic freeze-thaw action, and lead to the first-time formation of a deep active layer

in freshly deglaciated terrain (Hartmeyer et al. 2020).

Despite the intrinsically difficult comparison of geomorphic evidence from vastly different spatiotemporal scales, our findings indicate enhanced cirque growth during deglaciation. Two subsequent process-sets potentially underscore the geomorphic significance of deglaciation-related rockfall and the correlation between glacier retreat and cirque wall dismantling. Firstly, initial thermomechanical stresses due to deglaciation-induced rock mass damage condition instabilities that persist far beyond

the actual deglaciation (Graemiger et al., 2018); and secondly, the substantial input of fresh rockfall debris to the randkluft and into the sliding glacier base is likely to cause enhanced subglacial erosion and thus drive cirque erosion also over longer time scales (Sanders et al., 2013).

Rockwall retreat primarily advances along major joint intersections (Sass, 2005; Moore et al., 2009; Hartmeyer et al., 2020) and therefore rockfall activity varies considerably spatially across cirque walls (Fig. 5). The highest rates overall were recorded

at KN, the only of the five monitored rockwalls where inclination follows cleavage dip (~ 45 °N) and enables significant dip-slope failures of large cubic rock fragments (Fig. 2). High rates were also recorded at the Magnetkoepfl (MKW, MKE) where instability in part results from intense glacial strain during past glaciations due to the Magnetkoepfl's location between two ice flows (Fig. 1) providing intense abrasion.

During the six-year monitoring period headward cirque erosion at KN (10.3 mm a$^{-1}$ overall, 57.3 mm a$^{-1}$ in proximal areas)

exceeded lateral cirque erosion at KNW/MKE/MKW/MGE (0.9 mm a$^{-1}$ overall, 2.3 mm a$^{-1}$ in proximal areas) by an order of magnitude. At the western cirque (sidewalls MKW, MGE), only small remnants of a backwall exist, exhibiting a similar, cataclinal discontinuity setup as KN. While this specific area was not monitored in the present study, the absence of a significant backwall provides clear evidence for intense headward erosion in the past predisposed by the mica-schist cleavage.

The eastern cirque – confined by backwall KN and sidewalls KNW, MKE – and to a lesser degree the western cirque, display

clear north-south elongated shapes with long sidewalls (KNW, MGE) (Fig. 8). Length/width (L/W) ratios of both cirques (1.2 for western cirque, 1.5 for eastern cirque) exceed most L/W ratios given in the literature. In an extensive review of 25 different cirque inventories containing over 10,000 cirques from around the world, Barr and Spagnolo (2015) found a mean L/W ratio of 1.03 indicating that the majority of cirques is almost perfectly circular in shape. Deviations from circularity have mostly been attributed to glaciation history (review in Barr and Spagnolo, 2013) but geological structure is considered an important

factor, too: In the Western Alps and in the uplands of North Wales cirques that cut along geological strike are more elongated than those cutting across strike (Federici and Spagnolo, 2004; Bennett and Glasser, 2009), in the Carpathians the dipping of bedding planes significantly influences cirque shape (Mindrescu and Evans, 2014), and in the French Pyrenees bedrock characteristics are considered key controls of cirque shape in anisotropic schists (Delmas et al., 2015). Cirque morphology and lithology at the Kitzsteinhorn are in line with these findings and indicate that due to highly effective cleavage-driven sapping

the headward cirque erosion (north faces) has outpaced lateral cirque erosion (west and east faces) over extended time periods. The cataclinal backwalls thus not only affect short-term rockfall dynamics but also have a substantial effect on long-term erosion patterns that carved today's elongated cirque, which deviates significantly from ideal circularity.

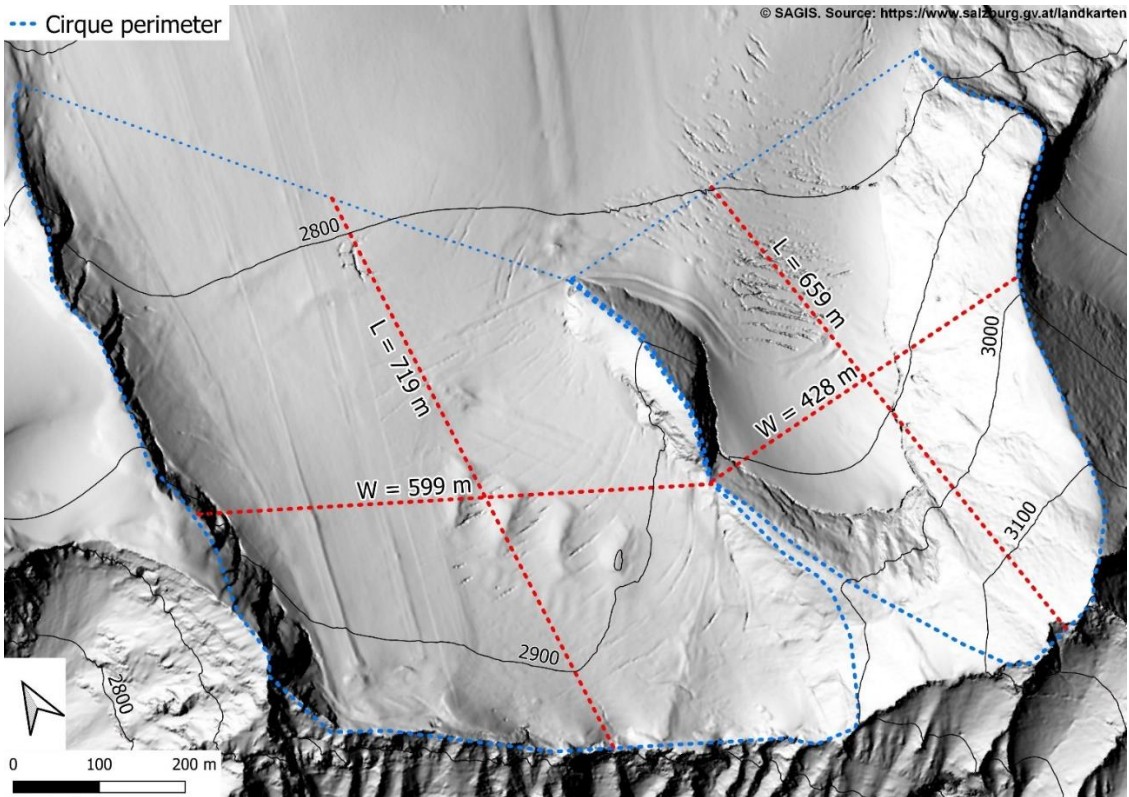

**Figure 8: Hillshade of the Kitzsteinhorn and the Schmiedingerkees glacier. Width (W) and length (L) of the two cirques indicated**
**(red dotted lines). Both cirques display a clear north-south elongated shape with high W/L ratios (1.2 for the western cirque, 1.5 for the eastern cirque) indicating effective cleavage-driven headward cirque erosion (north faces) over extended time periods.**

## 5.2 Rockfall Magnitudes and Frequencies

Many earlier studies have deduced magnitude-frequency distributions to statistically describe spatiotemporal rockfall variation in various non-glacial environments (Dussauge-Peisser et al., 2002). Here we present the first rockfall magnitude-frequency distribution for a deglaciating cirque (Fig. 7) that also shows characteristic negative power functions between rockfall number and rockfall volume (Hungr et al., 1999; Stark and Hovius, 2001).

During the monitoring period about two thirds of the rockwall retreat resulted from low-frequency rockfalls > 100 m³, while smaller high-frequency events being of minor relevance only (Fig. 3). At low magnitudes, no rollover (i.e. drop-off in rockfall frequency) was registered. Rollovers are more frequent in landslide inventories (Malamud et al., 2004) but have also been reported for rockfall inventories (Strunden et al., 2015). Their existence has been attributed to physical causes (Bennett et al., 2012), more frequently also as indicative of undersampling small failures due to insufficient spatial resolution (Stark and Hovius, 2001). The absence of a rollover in the inventory presented emphasizes consistent data quality and confirms an unbiased identification of rockfalls down to (at least) the cutoff threshold of 0.1 m³.

The contribution of rockfalls smaller than the specified size threshold of 0.1 m³ remains uncharacterized. Extending the power law established to the unspecified space below the 0.1 m³ cutoff threshold demonstrates that the unexamined small-magnitude rockfalls do not provide a significant contribution to the total mass wasting budget (magnitudes 0.001-0.1 m³ contribute around 10 m³ a$^{-1}$). Rock slope erosion in the two studied cirques is therefore dominated by the upper end of the investigated magnitude spectrum, which provides a valuable contribution to an ongoing debate on geomorphological effectiveness that dates back to the origins of quantitative research on rockwall retreat (Heim, 1932; Jaeckli, 1957).

Large rockfalls are rare in the observational record, yet they are of significant relevance for the shaping of alpine landscapes (Guthrie and Evans, 2007). How observations made over a few months or years relate to landscape evolution over millennia or longer (Paine, 1985) still remains elusive. The straight mathematical extrapolation (Sect. 4.4) indicates that rockfalls ≥ 10,000 m³ (≥ 100,000 m³) recur on multi-decadal (centennial) time scales. Data from short observation periods however, can only provide a partial explanation of episodic process dynamics (Crozier, 1999). Furthermore, event frequency estimates based on short records assume long-term stationary systems (e.g. Klemeš, 1993) – an assumption that is particularly unrealistic given the severity of recent climate change. Similar to hydrological systems, estimating the event probability for rare, large events outside the observed frequency-magnitude spectrum introduces significant time scale issues (Blöschl and Sivapalan, 1995) and should be considered with caution also in slope systems.

Due to the episodic dynamics of rockfall processes, sensitivity estimations of observed magnitude-frequency distributions such as performed in the present study (Sect. 4.4) are of high relevance. Repeated random removal of one fifth of the events (bootstrapping simulation) as well as the targeted removal of large events (> 100 m³) did not significantly alter the resulting power law exponents and thus confirms the validity of the differences observed between proximal and distal areas.

Values of the power law exponent *b* reported in the literature correlate negatively with lithological strength, as larger events are getting less frequent as slope strength decreases. Smaller exponents are typically observed in stronger bedrock, whereas

high values correspond either to slopes of weaker bedrock or to soil-mantled slopes (Bennett et al., 2012). Exponents for landslide inventories ($b \sim 1\text{-}1.5$) are therefore significantly higher than exponents for rockfall inventories, which typically range between 0.1-1 (Dussauge et al., 2003; Bennett et al., 2012). The power law exponent calculated here for distal rockwall sections ($b = 0.69$) falls in the upper range observed for rockfall distributions and clearly below the range determined for landslides. The high exponent is caused by (i) the rather high bedrock erodibility of the local calcareous micaschists (Terweh,

2012), and (ii) highly fractured zones with low lithological strength along structural weaknesses that are highly susceptible to rockfall (Hartmeyer et al., 2020; Sass, 2005; Moore et al., 2009).

Beside lithological strength, differences in power law exponents between different studies may also be related to the temporal data resolution (given here by the survey interval). Longer return periods between surveys increase the probability of superimposition and coalescence of events and result in decreasing failure numbers and increasing failure volumes, and thus

lower power law exponents (Barlow et al., 2012). Recent near-continuous LiDAR monitoring of cliff erosion highlights how different return periods between surveys affect magnitude-frequency distributions (Williams et al., 2018). While frequent monitoring provides more realistic magnitude-frequency distributions, longer durations between scans increase data precision and are advantageous for quantifying longer-term erosion rates due to reduced accumulative uncertainties (van Veen et al., 2017; Williams et al., 2018).

The significant differences between magnitude-frequency distributions in proximal areas ($b = 0.51$) and distal areas ($b = 0.69$) cannot be related to variations in lithology, sampling interval, or stochastic behaviour but result from the destabilizing effect of recent ice retreat. Following deglaciation, pronounced thermal stress is induced and an active layer penetrates into the exposed bedrock (Hartmeyer et al., 2020; Draebing and Krautblatter, 2019) contributing to an increased rockfall frequency close to the glacier surface. Variations between rockfall magnitude-frequency distributions in (proximal) areas directly affected

by deglaciation and (distal) areas unaffected by recent deglaciation become most apparent at the upper end of the magnitude spectrum investigated: The (normalized) frequency of rockfalls > 100 m³ is almost ten times higher in proximal areas than in distal areas (Fig. 6). The increased occurrence of large rockfalls in the vicinity of the glacier leads to a flatter regression line and thus to a reduced power law exponent for proximal areas. In distal areas large rockfalls are of reduced importance and the rockfall volume distribution here shows a less pronounced rise and peaks at lower magnitudes (Fig. 4b), which could

theoretically contribute to an 'archway-shaped' pattern proposed for hard metamorphic rocks with a single maximum of mid-magnitude rockfalls (Krautblatter et al., 2012).

Substituting space for time, the results indicate how recent climate warming modifies spatiotemporal rockfall occurrence in glacial environments over decades subsequent to glacier downwasting. The patterns obtained are important for rockfall hazard assessments as they indicate that in rockwalls affected by glacier retreat historical rockfall patterns may no longer be used as

indicators for future events (Sass and Oberlechner, 2012; Krautblatter and Moore, 2014). Recurrence intervals of different events are proposed to change with time since deglaciation, which is helpful information for sediment cascade modelling (e.g. Bennett et al., 2014) and hazard assessment more generally. The latter is particularly critical in cirque environments, where the presence of glacially oversteepened rockwalls and low-friction glacier surfaces promote long rockfall runout distances

(Schober et al., 2012). Many high-alpine, glacier tourism areas that are enjoying growing popularity worldwide, may have to adapt risk-reduction measures in the near future (Purdie et al., 2015).

## 6 Conclusions

- A unique six-year rockfall inventory (2011-2017) from the lateral and back-walls of two elongated glaciated cirques in the Hohe Tauern Range, Central European Alps, Austria provides unprecedented insights into rockfall dynamics in deglaciating terrain. The inventory was derived from detailed terrestrial LiDAR data and represents the most extensive high-resolution compilation of rockfall in cirque walls. Details on site-specific deglaciation, spatial rockfall distribution and potential causes behind the observed rockfall increase in freshly deglaciated terrain are discussed in a companion study (Hartmeyer et al., 2020). In the present paper we analysed cirque wall retreat and magnitude-frequency distributions and draw the following conclusions: High mean cirque wall retreat of 1.9 mm a$^{-1}$ ranking in the top of reported values worldwide;

- Pre-existing structural weaknesses modify spatial patterns of rockfall activity;

- For glacier-proximal (0-10 m above glacier surface) areas mean retreat rates are an order of magnitude higher than, for distal areas (> 10 m);

- Enhanced cirque wall dismantling in recently deglaciated rockwall sections supports concepts of increased cirque growth during deglacial periods;

- Elongated cirque morphology fits to the pattern in cirque wall retreat rates indicating that headward erosion (10.3 mm a$^{-1}$ at KN) outpaces lateral erosion (0.9 mm a$^{-1}$ combined mean for all sidewalls) significantly. The essential factor is the direction of cleavage so that in cataclinal backwalls large dip-slope failures are facilitated and drive effective headward sapping;

- The rockfall magnitude-frequency distribution in the deglaciating cirque follows a clear negative power law distribution over four orders of magnitude (b = 0.64);

- Rockfall magnitude-frequency distributions in glacier-proximal areas (b = 0.51) and distal areas (b = 0.69) differ significantly and demonstrate a greater importance of large rockfalls in recently deglaciated rockwalls;

- Climate warming-induced glacier retreat enhances rockfall occurrence in cirque walls; a persistent pattern that will likely continue until after glaciers have fully disappeared.

**Data availability.** The rockfall inventory can be downloaded from the *mediaTUM* data repository under the following weblink: https://mediatum.ub.tum.de/1540134.

**Author contributions.** MKE, LS and JO initiated the underlying research project in 2010 and obtained the funding. IH, MKE and RD developed the idea and designed the study. IH and RD conducted the data acquisition and the data analysis. GP

performed the statistical computing. All authors contributed to the discussion and interpretation of the data. IH drafted the manuscript with significant contributions from MKR and AL.

**Competing interests.** The authors declare that they have no conflict of interest.

**Acknowledgements**. We would like to thank Georgina Bennett and Arjun Heimsath, and Associate Editor Arjen Stroeven for their thoughtful feedback and constructive reviews. A. Schober kindly provided the photographs used for Fig. 2.

**Financial support.** This study was co-funded by the Austrian Academy of Sciences (ÖAW) (Project 'GlacierRocks'), the Arbeitsgemeinschaft Alpenländer (ARGE ALP) (Project 'CirqueMonHT') and the Austrian Research Promotion Agency (FFG) (Project 'MOREXPERT'). We furthermore thank the Gletscherbahnen Kaprun AG (Project 'Open-Air-Lab Kitzsteinhorn') for financial and logistical support.

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
