# Peer review of "6-year LiDAR survey reveals enhanced rockwall retreat and modified rockfall magnitudes/frequencies in deglaciating cirques"

_Earth Surface Dynamics, 2020_

## Referee Comment (RC1) · Georgina Bennett (Referee) · 26 Mar 2020

This study presents an inventory of rockfall events on cirque walls in an alpine landscape in Austria collected using repeat laser scanning. The authors find higher rates on the back wall due to fracture orientation and find that these are some of the highest rates of cirque retreat reported anywhere globally. They produce the first magnitude frequency distributions for cirque rockfall and find some interesting differences with distance from the glacier surface. Most significantly, rockfalls closer to the glacier surface, i.e. more recently deglaciated, tend to be larger, resulting in a

lower power law exponent in the magnitude frequency relationship. This study shows how rockfall hazard evolves following deglaciation forming valuable input to sediment cascade models and for hazard assessment in these landscapes. I really enjoyed reading this paper, which presents a very interesting dataset in a clear and generally well-presented way, and does a good job of highlighting the novelty of the research. It also does a very good job of covering the main literature on the topic, though I am better placed to say this from the perspective of magnitude frequency distributions of landslides/rockfall, rather than of cirque geomorphology, which I have less knowledge of. As a researcher interested in alpine sediment cascades, I would say that the main contribution of the paper is the quantification of how magnitude frequency distributions of rockfall evolve following deglaciation. However, I would suggest that the paper would be of appeal to geomorphologists interested in the role of geological structure on rockfall characteristics as well as those interested in glacial geomorphology. I think it is well placed in this journal following some minor revisions. I have made some more comments on the PDF attached but some points I noted down during the review are: Sometimes it is difficult to distinguish results presented in the two different papers. Perhaps make it clear briefly in the introduction the key findings and differences between these. You emphasize differences in the rates of cirque retreat in this paper but need to visualize how your rates compare to others and why they might differ. You mention that it is hard to compare rates over different spatial and temporal scales. Perhaps you could produce a figure or table comparing rates collected over similar spatial and temporal scales and perhaps a bit more discussion on why these might differ and why yours are so high. You seem to find a more elongated cirque geometry than the normal due to the effect of cataclinal back walls on erosion rates. Again, it would be interesting to see a plot visualizing the shape of this cirque in comparison to others, perhaps as a function of geological structure wherever this information is available? There is some information on uncertainties in the power law distributions e.g. resulting from inclusion or not of the largest rockfalls, that should be presented in the results section and relevant figure. You also lack quantification of error in your rockfall volumes i.e. the propagation of alignment errors between 1-2 cm. These should ideally be propagated into volume error following the method used in Bennett et al. 2012. All the best with your revisions, Georgie Bennett

Please also note the supplement to this comment:
https://www.earth-surf-dynam-discuss.net/esurf-2020-9/esurf-2020-9-RC1-supplement.pdf

**Supplement:**

[revised manuscript text omitted]

---

## Author Comment (AC1) · 27 Mar 2020

We thank Georgina Bennett for her positive and constructive comments and are happy to include the suggested amendments in the revision.
* * *

---

## Referee Comment (RC2) · Arjun Heimsath (Referee) · 7 Apr 2020

I greatly enjoyed reading this paper, especially the well referenced introduction that reminded me of the wide-ranging aspects of this challenging problem – quantifying rockwall retreat and erosion rates of headwalls above glaciers. The approach taken here is innovative and interesting and the results presented are compelling, though perhaps not as robust and broadly applicable as the authors assert. I believe that with relatively easy edits this will be a widely cited paper in studies tackling hazard implications of climate change as well studies tackling how quickly bedrock faces erode.

[Figure]

The journal is suitable and the paper is in pretty good shape.

A few suggestions that will make it a stronger contribution, I believe.

Firstly, this is a study compiling 6 years' worth of repeat LIDAR scans of the same glacial features. This needs to be explicitly addressed and acknowledged in the discussion of results as well as in the introduction. In the introduction, I suggest including a section distinguishing between long term average studies (e.g. ones that use cosmogenic nuclides in one way or another like Greg Stock's extensive work in Yosemite or the approach of Heimsath and McGlynn (Geomorphology, 2008)) and the shorter time scale ones that are reporting results from monitoring studies such as this one (note that the Alaska paper of O'Farrell, Heimsath et al. (ESPL 07) had a short section of converting scree deposit to rock retreat rates as an example of short term studies). In the discussion section, it would be helpful to have a more thorough examination of the inferred frequency magnitude curves given limits in the data. Extensive examples from the hydrological sciences address the time scale issue and perhaps some can be used as template for framing this discussion (apologies, while I remember reading such papers I don't remember who they were by – I was better versed in this literature years ago).

Second, I think a conceptual sketch/model to accompany Figure 2 would help a lot for visualizing how the authors tackle this problem. I found Figure 2 almost incomprehensible and if it had a sketch accompanying it that showed how measurements made in this study resulted in the inference of a retreat rate that would be great.

To this end, there really needs to be a better explanation of how these data are used to infer headwall retreat rates. Which areas were used and how exactly were the calculations done? What's the uncertainty on those calculations and what are the assumptions and simplifications? All of these questions could be illustrated in some way in a good conceptual model.

Similarly, I think there needs to be better justification for the log binning approach.

Given the short methods sentence addressing this I have no way to evaluate how good it is and whether it is justified. Does it introduce some bias? Convince me better that it does not with more analyses. The key question is whether it would make the reporting of results based on percentage of total rockfall volume quite different depending on how the sizes were binned?

Finally, the distinctions between this paper and its companion paper could be made more clearly.

Let me know if I can help clarify any of these points and I hope the above is helpful.

Arjun Heimsath

---

## Author Comment (AC2) · 12 Apr 2020

We thank Arjun Heimsath for his thoughtful comments and his efforts towards improving our manuscript. We will include his suggestions into a revised version of the manuscript.

---

## Author Comment (AC5) · 19 May 2020

Dear reviewers, Dear editor,

thank you for your highly valuable comments. We posted point-by-point responses to your comments and uploaded the revised manuscript (one version with markups showing revisions, the other with a clean layout).

Following the comments by Arjun Heimsath (RC2) and comments made by referees of the companion paper (esurf-2020-8) the sensitivity analysis was extended and a bootstrapping exercise included. Günther Prasicek carried out the bootstrapping simulation and was added as co-author.

Kind regards

Ingo Hartmeyer (on behalf of all co-authors)

---

## Author Response (AR1)

**Response to RC1 (Georgina Bennett):**

Dear Georgina Bennett,

thank you for your constructive and insightful comments which we feel helped to improve the manuscript.

We (i) uploaded a revised version of the manuscript (changes highlighted and commented), (ii) posted a new author's comment to inform on minor general amendments in the manuscript, and (iii) below provide a point-by-point response to all your comments. Our response is structured as follows: (1) referee comment, (2) author's response and (3) changes in manuscript text.

(1) Sometimes it is difficult to distinguish results presented in the two different papers. Perhaps make it clear briefly in the introduction the key findings and differences between these.

(2) To highlight the distinctions of the companion papers we (i) expanded the abstract, (ii) summarized the key findings of the companion paper in the introduction (Sect. 1), and (iii) slightly modified the conclusions (Sect. 6).

(3) The summary given in the introduction reads as follows:
This paper is closely linked to a companion paper (Hartmeyer et al., 2020) which identified significant glacial thinning (0.5 m a$^{-1}$) adjacent to the monitored rockwalls and found elevated rockfall activity in the freshly deglaciated terrain. 60 % of the rockfall volume detached from less than ten vertical meters above the glacier surface. High rates 10-20 m above the glacier indicate enhanced rockfall activity over tens of years following deglaciation. Rockfall preconditioning probably starts inside the Randkluft (void between cirque wall and glacier) where sustained freezing and ample supply of liquid water causes enhanced physical weathering and high plucking stresses. As the glacier is wasting down strong temperature variations will induce pronounced thermal stress in the first-time exposed rock, cause rock fatigue and lead to the formation of a deep active layer, all of which will exert significant destabilizing effects in glacier-proximal areas.

###

(1) You emphasize differences in the rates of cirque retreat in this paper but need to visualize how your rates compare to others and why they might differ. You mention that it is hard to compare rates over different spatial and temporal scales. Perhaps you could produce a figure or table comparing rates collected over similar spatial and temporal scales and perhaps a bit more discussion on why these might differ and why yours are so high.

(2) We added a table listing cirque wall retreat rates from various studies and tried better distinguishing between retreat rates from cirques and from other arctic/alpine environments in the text. We also expanded the text to include potential causes for the elevated retreat rates observed.

###

(1) You seem to find a more elongated cirque geometry than the normal due to the effect of cataclinal back walls on erosion rates. Again, it would be interesting to see a plot visualizing the shape of this cirque in comparison to others, perhaps as a function of geological structure wherever this information is available?

(2) Based on literature data we compared length/width ratios of cirques investigated here and those of other studies. We also added a plot visualizing the elongated shape of the monitored cirques and extended the review on structural controls of cirque shape.

(3) **The text now reads:** Length/width (L/W) ratios of both cirques (1.5 for eastern cirque, 1.2 for western cirque) exceed most L/W ratios given in the literature. In an extensive review of 25 different cirque inventories containing over 10,000 cirques from around the world, Barr and Spagnolo (2015) found a mean L/W ratio of 1.03 indicating that the majority of cirques is almost perfectly circular in shape. Deviations from circularity have mostly been attributed to glaciation history (review in Barr and Spagnolo, 2013) but geological structure is considered an important factor, too: In the Western Alps and in the uplands of North Wales cirques that cut along geological strike are more elongated

than those cutting across strike (Federici and Spagnolo, 2004; Bennett and Glasser, 2009), in the Carpathians the dipping of bedding planes significantly influences cirque shape (Mindrescu and Evans, 2014), and in the French Pyrenees bedrock characteristics are considered key controls of cirque shape in anisotropic schists (Delmas et al., 2015).

###

(1) There is some information on uncertainties in the power law distributions e.g. resulting from inclusion or not of the largest rockfalls, that should be presented in the results section and relevant figure.
(2) The respective paragraphs have been transferred to the results section (Sect 4.4).

###

(1) You also lack quantification of error in your rockfall volumes i.e. the propagation of alignment errors between 1-2 cm. These should ideally be propagated into volume error following the method used in Bennett et al. 2012.
(2) Rockfall volume errors are discussed in the companion paper. We modified the text and added a cross-reference. Based on your suggestion the alignment error has now been integrated.
(3) For each distance measurement the algorithm calculates the local confidence interval (at one sigma level), which was added to the alignment error and propagated into the volume error. Full details on rockfall volume computation and error quantification are provided in Hartmeyer et al., 2020. In addition to rockfall volume and its associated uncertainty a suite of morphometric parameters including slope aspect, gradient and elevation above glacier surface was determined for each rockfall source area.

**Response to RC2 (Arjun Heimsath):**

Dear Arjun. Heimsath,

thank you for your constructive and insightful comments which we feel helped to improve the manuscript.

We (i) uploaded a revised version of the manuscript (changes highlighted), (ii) posted a new author's comment to inform on minor general amendments in the manuscript, and (iii) below provide a point-by-point response to all your comments. Our response is structured as follows: (1) referee comment, (2) author's response and (3) changes in manuscript text.

(1) Firstly, this is a study compiling 6 years' worth of repeat LIDAR scans of the same glacial features. This needs to be explicitly addressed and acknowledged in the discussion of results as well as in the introduction.
(2) Has now been made more explicit in (i) introduction, (ii) results, and (iii) discussion sections.

###

(1) In the introduction, I suggest including a section distinguishing between long term average studies (e.g. ones that use cosmogenic nuclides in one way or another like Greg Stock's extensive work in Yosemite or the approach of Heimsath and McGlynn (Geomorphology, 2008)) and the shorter time scale ones that are reporting results from monitoring studies such as this one (note that the Alaska paper of O'Farrell, Heimsath et al. (ESPL 07) had a short section of converting scree deposit to rock retreat rates as an example of short term studies).
(2) We included a section distinguishing between short- and long-term studies to the introduction and integrated both suggested references (O'Farrell et al., 2009; Heimsath and McGlynn, 2008). We also included a table to the discussion (Table 1) that provides the time scale of the listed investigations.
(3) **The text now reads:** Cirque wall retreat rates have been quantified using a variety of different approaches including long-term averages based on sediment deposits (e.g. Larsen and Mangerud, 1981), cosmogenic dating (e.g. Heimsath and McGlynn, 2008), and cirque allometry (e.g. Evans et al., 2006) as well as short-term monitoring studies based on lacustrine deposits (e.g. Hicks et al., 1990), supraglacial scree (e.g. O'Farrell et al. 2009), and terrestrial LiDAR (e.g. Kenner et al., 2011).

###

(1) In the discussion section, it would be helpful to have a more thorough examination of the inferred frequency magnitude curves given limits in the data. Extensive examples from the hydrological sciences address the time scale issue and perhaps some can be used as template for framing this discussion (apologies, while I remember reading such papers I don't remember who they were by – I was better versed in this literature years ago).
(2) We added a new paragraph to the discussion (Sect. 5.2) that addresses time scale issues and limits of the frequency-magnitude concept. Utilizing a bootstrapping simulation we further constrained the robustness of the sensitivity analysis inferred from the magnitude-frequency distribution.
(3) **The text now reads:** Large rockfalls are rare in the observational record, yet they are of significant relevance for the shaping of alpine landscapes (Guthrie and Evans, 2007). How observations made over a few months or years relate to landscape evolution over millennia or longer (Paine, 1985) still remains elusive. The straight mathematical extrapolation (Sect. 4.4) indicates that rockfalls $\geq 10,000$ m³ ($\geq 100,000$ m³) recur on multi-decadal (centennial) time scales. Data from short observation periods however, can only provide a partial explanation of episodic process dynamics (Crozier, 1999). Furthermore, event frequency estimates based on short records assume long-term stationary systems (e.g. Klemeš, 1993) – an assumption that is particularly unrealistic given the severity of recent climate change. Similar to hydrological systems, estimating the event probability for rare, large events outside the observed frequency-magnitude spectrum introduces significant time scale issues (Blöschl and Sivapalan, 1995) and should be considered with caution also in slope systems.

Due to the episodic dynamics of rockfall processes, sensitivity estimations of observed magnitude-frequency distributions such as performed in the present study (Sect. 4.4) are of high relevance. Repeated random removal of one fifth of the events (bootstrapping simulation) as well as the targeted removal of large events (> 100 m³) did not significantly alter the resulting power law exponents and thus confirms the validity of the differences observed between proximal and distal areas.

###

(1) Second, I think a conceptual sketch/model to accompany Figure 2 would help a lot for visualizing how the authors tackle this problem. I found Figure 2 almost incomprehensible and if it had a sketch accompanying it that showed how measurements made in this study resulted in the inference of a retreat rate that would be great.
(2) We adapted Figure 2 and hope that in combination with the additional section (Sect. 3.3) on the calculation of rockwall retreat rates the reader now gets a better understanding of the geological structure and rockwall retreat rates inferred.

###

(1) To this end, there really needs to be a better explanation of how these data are used to infer headwall retreat rates. Which areas were used and how exactly were the calculations done? What's the uncertainty on those calculations and what are the assumptions and simplifications? All of these questions could be illustrated in some way in a good conceptual model.
(2) We added a new subchapter (Sect. 3.3) to describe the calculation of rockwall retreat rates (and modified Fig. 2). We also added the uncertainty of the retreat rates to the text and the figure in Sect. 4.2 (Figure 5 now contains error bars).
**(3) The text now reads:**
3.3 Rockwall Retreat Rate Calculation
First, the total rockfall volume registered was divided by the number of observation years to obtain mean annual volume (m³). Second, the volume was divided by the surface area of the investigated rockwall (m²) to derive the (slope-perpendicular) rockwall retreat rate. Rockwall surface area calculations were carried out in CloudCompare: point clouds of rockwalls were first subsampled (thinned) to a homogenous point density of 0.5 m$^{-1}$ in order to prevent a potential bias due to variable resolution within point clouds. The subsampled point cloud was then used to generate a mesh based on a Delaunay triangulation (maximum edge length 4 m), which served as basis for the surface area calculation.

###

(1) Similarly, I think there needs to be better justification for the log binning approach. Given the short methods sentence addressing this I have no way to evaluate how good it is and whether it is justified. Does it introduce some bias? Convince me better that it does not with more analyses. The key question is whether it would make the reporting of results based on percentage of total rockfall volume quite different depending on how the sizes were binned?
(2) We added a paragraph to justify the log binning approach.
(3) **The text now reads:** Magnitude-frequency distributions of rockfalls often follow a power law function (as is also the case here, see Sect. 4.4). To classify rockfall volumes, the recorded events were grouped into bins of logarithmically increasing size to balance against strongly uneven event volumes (Fig. 3). This follows the volumetric classification introduced by Whalley (1974, 1984) (debris falls < 10 m³; boulder falls $10-10^2$ m³; block falls $10^2-10^4$ m³), which is commonly used in science and engineering (e.g. Brunetti et al., 2009; Krautblatter et al., 2012; Sellmeier, 2015).

###

(1) Finally, the distinctions between this paper and its companion paper could be made more clearly.
(2) To highlight the distinctions of the companion papers we (i) expanded the abstract, (ii) summarized the key findings of the companion paper in the introduction (Sect. 1), and (iii) modified the conclusions (Sect. 6).

(3)   The summary given in the introduction reads as follows:

[revised manuscript text omitted]

**Kommentiert [IH3]:** Modified following suggestion by RC2 (A. Heimsath):

RC2 (A. Heimsath) originally wrote:
*"In the introduction, I suggest including a section distinguishing between long term average studies (e.g. ones that use cosmogenic nuclides in one way or another like Greg Stock's extensive work in Yosemite or the approach of Heimsath and McGlynn (Geomorphology, 2008)) and the shorter time scale ones that are reporting results from monitoring studies such as this one (note that the Alaska paper of O'Farrell, Heimsath et al. (ESPL 07) had a short section of converting scree deposit to rock retreat rates as an example of short term studies)."*

[revised manuscript text omitted]

**Kommentiert [IH7]:** Both reviewers (RC1, RC2) suggested clearer distinctions between the companion studies.

RC1 (G. Bennett) originally wrote:
*"Sometimes it is difficult to distinguish results presented in the two different papers. Perhaps make it clear briefly in the introduction the key findings and differences between these."*

**with monitored rockwalls (blue) and glacier extent. 1c: Location of study area within Austria. Abbreviations: K = Kitzsteinhorn (Summit), SMK = Scan Position 'Magnetkoepfl', SCC = Scan Position 'Cable Car Top Station', SG1 = Scan Position 'Glacier 1', SG2 = Scan Position 'Glacier 2', SMG = Scan Position 'Maurergrat', see text for further abbreviations (Photo: UAV/R. Delleske).**

The Schmiedingerkees glacier and the immediately adjacent summit pyramid of the Kitzsteinhorn constitute one of Austria's most frequented high-alpine tourist destinations with close to one million visitors per year. The Kitzsteinhorn hosts an extensive research site to investigate the consequences of climate change on high-alpine infrastructure and rock stability ('Open-Air-Lab Kitzsteinhorn'). Measurements performed at the Kitzsteinhorn focus on rockfall activity (Keuschnig et al., 2015), subsurface temperature changes (Hartmeyer et al., 2012), geophysical monitoring with ERT (Supper et al., 2014; Keuschnig et al., 2016), rock mass pressure using anchor load plates (Plaesken et al., 2017) and fracture dynamics monitoring with crackmeters (Ewald et al., 2019).

Full details of the study site are presented in Hartmeyer et al., 2020submitted. In brief, the total surface area of the investigated rockwalls is 234,700 m² and their mean vertical extent ranges between 35-200 m. All studied cirque walls are immediately adjacent to the Schmiedingerkees glacier: In the eastern cirque the Kitzsteinhorn north-face (KN) (backwall), Kitzsteinhorn northwest-face (KNW) and Magnetkoepfl east-face (MKE) (sidewalls). In the western cirque no significant backwall exists and the Magnetkoepfl west-face (MKW) and Maurergrat east-face (MGE) were monitored (Fig. 1).

All cirque walls developed in rocks of the Bündner schist formation within the Glockner Nappe and belong to the Glockner Facies consisting of calcareous micaschist, prasinite, amphibolite, phyllite, marble and serpentinite (Cornelius and Clar, 1935; Hoeck et al., 1994). Within the monitored rockwalls, NNE dipping (~ 45 °) calcareous micaschists dominate, and isolated marble and serpentinite belts exist at Magnetkoepfl. Two distinct joint sets (J1 dipping steep to W, J2 medium-steep to SW) oriented approximately orthogonal to the cleavage precondition disintegration into cubic rock fragments (Fig. 2). Numerous open fractures infilled with fine-grained material enable water infiltration and affect near-surface rock slope kinematics and thermal dynamics (Keuschnig et al., 2016; Ewald et al., 2019). Rock at the surface is highly fractured due to a pronounced frost weathering susceptibility. Strong tectonic forcing resulted in highly fractured weakness zones along major faults in all rockwalls. Rock mass classifications according to Romana, 1985 and Bieniawski, 1993 suggest highly variable lithological strength ranging from low stability values in weakened zones (r = 34) and highly stable conditions in steep, unweathered sections (r = 98) (Terweh, 2012).

[Figure]

**Figure 2:**  Dip and strike of the Ddominant discontinuit ies at the studied rockwalls (Fig. 2A). (a & b)  Images of Kitzsteinhorn north-face (KN) (Fig. 2B/C) and  Magnetkoepfl east-face (MKE) (Fig. 2D) . with Ccleavage (CL) and joint sets (J1, J2) indicated. Cleavage of  calcareous mica-schists dips about 45° NNE. Joint sets J1 (dipping subvertical to W) and J2 (dipping steeply to SW) are approximately orthogonal to CL and predispose north-facing slopes for dip-slope failures (Fig. 2B/C), while west- and east-facing areas are more susceptible to toppling failures (Fig. 2D) (Photos: R. Delleske (2B). A.  Schober (2C, 2D)).

**Kommentiert [IH8]:** Figure 2 modified following suggestion by RC2 (A. Heimsath):

RC2 (A. Heimsath) originally wrote:
*"I found Figure 2 almost incomprehensible and if it had a sketch accompanying it that showed how measurements made in this study resulted in the inference of a retreat rate that would be great."*

[revised manuscript text omitted]

**Kommentiert [IH10]:** Sensitivity analysis was expanded following suggestions by RC2 (A. Heimsath) and reviewers of the companion paper.

**Kommentiert [IH11]:** This subchapter was added following suggestion by RC2 (A. Heimsath).

RC2 (A. Heimsath) originally wrote:
*"there really needs to be a better explanation of how these data are used to infer headwall retreat rates. Which areas were used and how exactly were the calculations done? What's the uncertainty on those calculations and what are the assumptions and simplifications?"*

**Kommentiert [IH12]:** Modified following RC2 (A. Heimsath):

RC2 (A. Heimsath) originally wrote:
*"this is a study compiling 6 years' worth of repeat LIDAR scans of the same glacial features. This needs to be explicitly addressed and acknowledged in the discussion of results as well as in the introduction."*

[revised manuscript text omitted]

**Kommentiert [IH23]:** Modification following RC1 (G. Bennett)

RC1 (G. Bennett) originally wrote:
"*You emphasize differences in the rates of cirque retreat in this paper but need to visualize how your rates compare to others and why they might differ. You mention that it is hard to compare rates over different spatial and temporal scales. Perhaps you could produce a figure or table comparing rates collected over similar spatial and temporal scales and perhaps a bit more discussion on why these might differ and why yours are so high.*"

**Kommentiert [IH24]:** Modified following suggestion by RC1 (G. Bennett)

RC1 (G. Bennett) originally wrote:
"*You emphasize differences in the rates of cirque retreat in this paper but need to visualize how your rates compare to others and why they might differ. You mention that it is hard to compare rates over different spatial and temporal scales. Perhaps you could produce a figure or table comparing rates collected over similar spatial and temporal scales and perhaps a bit more discussion on why these might differ and why yours are so high.*"

[revised manuscript text omitted]

**Kommentiert [IH31]:** Modification following RC1 (G. Bennett).

RC1 (G. Bennett) originally wrote:
*"This needs a bit more explanation."*

**Kommentiert [IH32]:** Modification following RC1 (G. Bennett).

RC1 (G. Bennett) originally wrote:
*"I would not say that you clearly show this. You indirectly through space for time substitution, how [sic] how climate warming may alter rockfall characteristics through glacier downwasting."*

**Kommentiert [IH33]:** Modified following RC1 (G. Bennett).

RC1 (G. Bennett) originally wrote:
*"Because they are biased towards larger events?*

*I think your results show how rockfall characteristics, namely magnitude frequency statistics, change with time since deglaciation. You may therefore calculate how the recurrence intervals of different events may alter with time since deglaciation, which is very helpful information for sediment cascade modeling (e.g. Bennett et al., 2014) and hazard assessments more generally."*

670 resolution compilation of rockfall in cirque walls. Details on site-specific deglaciation, spatial rockfall distribution and potential causes behind the observed rockfall increase in freshly deglaciated terrain are discussed in a companion study (Hartmeyer et al., 2020). In the present paper we analysed cirque wall retreat and magnitude-frequency distributions and draw the following conclusions: Besides significantly increased rockfall occurrence close to the glacier surface due to rockfall preconditioning inside the Randkluft and ice surface lowering (Hartmeyer et al., submitted), the analysis of cirque wall retreat

675 and rockfall magnitude-frequency distributions clearly indicates:

- High mean cirque wall retreat of 1.9 mm a$^{-1}$ ranking in the top of reported values worldwide;
- Rockfall activity is boosted in the freshly exposed rock surfaces above the glacier;
- Pre-existing structural weaknesses modify spatial patterns of rockfall activity;
- For glacier-proximal (0-10 m above glacier surface) areas mean retreat rates are an order of magnitude higher than, for distal areas (> 10 m);

680
- Enhanced cirque wall dismantling in recently deglaciated rockwall sections supports concepts of increased cirque growth during deglacial periods;
- Elongated cirque morphology fits to the pattern in cirque wall retreat rates indicating that headward erosion (10.34 mm a$^{-1}$ at KN) outpaces lateral erosion

685
- (0.9 mm a$^{-1}$ combined mean for all sidewalls) significantly. The essential factor is the direction of cleavage so that in cataclinal backwalls large dip-slope failures are facilitated and drive effective headward sapping;
- The rockfall magnitude-frequency distribution in the deglaciating cirque follows a clear negative power law distribution over four orders of magnitude (b = 0.64);
- Rockfall magnitude-frequency distributions in glacier-proximal areas (b = 0.51) and distal areas (b = 0.69) differ

690 significantly and demonstrate a greater importance of large rockfalls in recently deglaciated rockwalls;
- Climate warming-induced glacier retreat enhances rockfall occurrence in cirque wallsglacial environments; a persistent pattern that will likely continue until after glaciers have fully disappeared.

**Data availability.** The rockfall inventory can be downloaded from the *mediaTUM* data repository under the following weblink: https://mediatum.ub.tum.de/1540134.

695 **Author contributions.** MKE, LS and JO initiated the underlying research project in 2010 and obtained the funding. IHMKE, IHMKE and RD developed the idea and designed the study. IH and RD conducted the data acquisition and the data analysis. GP performed the statistical computing. All authors contributed to the discussion and interpretation of the data. IH drafted the manuscript with significant contributions from MKR and AL.

**Competing interests.** The authors declare that they have no conflict of interest.

Kommentiert [IH34]: Both reviewers (RC1, RC2) suggested clearer distinctions between the companion studies.

Kommentiert [IH35]: As suggested by RC1 (G. Bennett)

[revised manuscript text omitted]

---

## Author Response (AR2)

Dear Editors,

thank you very much for your excellent feedback. We implemented all requested technical corrections and adapted the title.
5    Below you find a marked-up manuscript version.

Kind regards
Ingo Hartmeyer (on behalf of all co-authors)

[revised manuscript text omitted]